# Influence of altitude on cerebral and splanchnic oxygen saturation in critically ill children during air ambulance transport

Tova Hannegård Hamrin[1,2]*, Staffan Eksborg[1,3], Jonas Berner[1,2], Urban Fläring[1,2], Peter J. Radell[1,2]

1 Pediatric Perioperative Medicine and Intensive Care, Astrid Lindgren Children's Hospital, Karolinska University Hospital Solna, Stockholm, Sweden, 2 Department of Physiology and Pharmacology, Section of Anesthesiology and Intensive Care, Karolinska Institutet, Astrid Lindgren Children's Hospital, Karolinska University Hospital Solna, Stockholm, Sweden, 3 Childhood Cancer Research Unit, Department of Women's and Children's Health, Karolinska Institutet, Astrid Lindgren Children's Hospital, Barnläkemedelsgruppen, Norrbacka S3:04, Karolinska University Hospital Solna, Stockholm, Sweden

* tova.hannegard-hamrin@sll.se

**Data Availability Statement:** All relevant data are within the manuscript and its Supporting Information files.

## Abstract

### Objective

The aim of the current study was to investigate how cerebral and splanchnic oxygen saturation ($rSO_2$-C and $rSO_2$-A) in critically ill children transported in air ambulance was affected by flight with cabin pressurization corresponding to $\geq 5000$ feet. A second aim was to investigate any differences between cyanotic and non-cyanotic children in relation to cerebral and splanchnic oxygen saturation during flight $\geq 5000$ feet. The variability of the cerebral and splanchnic Near Infrared Spectroscopy (NIRS) sensors was evaluated.

### Design

NIRS was used to measure $rSO_2$-C and $rSO_2$-A during transport of critically ill children in air ambulance. $rSO_2$ data was collected and stored by the NIRS monitor and extracted and analyzed off-line after the transport. Prior to evaluation of the NIRS signals all zero and floor-effect values were removed.

### Setting

The Pediatric Intensive Care Unit (PICU) at Astrid Lindgren Children's Hospital, Karolinska University Hospital in Stockholm, Sweden.

### Patients

In total, 44 critically ill children scheduled for inter-hospital transport by a specialized pediatric transport team were included in the study between January 2014 and January 2019 (convenience sampling).

**Funding:** Financial support from the regional agreement on medical training and clinical research (ALF), project 20150224 between Stockholm County Council and Karolinska Institutet (Dr S Eksborg), Tova Hannegård Hamrin (THH) reported grants from Samariten Foundation: reference id 2017-0313 and the Swedish Society of Pediatric Anesthesiology and Intensive Care. The funders had no role in study design, data collection and analysis, decision to publish, or preparation of the manuscript.

**Competing interests:** The authors have declared that no competing interests exist.

### Intervention

No interventions were conducted.

### Measurements

All study patients were monitored with a cerebral NIRS-sensor placed over the forehead and an abdominal NIRS-sensor placed in the infra-umbilical area for cerebral and splanchnic regional oxygen saturation monitoring, $rSO_2$-C and $rSO_2$-A, respectively.

### Main results

Complete $rSO_2$-C and $rSO_2$-A data was obtained in 39 patients. Median age was 12 days. Cyanotic congenital heart malformations were present in 9 patients (23%). In 22 patients (56%) $rSO_2$-C decreased at altitude $\geq$ 5000 feet and in 24 patients (61%) $rSO_2$-A decreased at altitude $\geq$ 5000 feet compared to baseline (p<0.0001). In 25 patients (64%) the $rSO_2$-C/$rSO_2$-A ratio was greater at altitude $\geq$ 5000 feet than at baseline. A ratio $\geq$ 1 was seen in 77% of patients at altitude $\geq$ 5000 feet compared to in 67% of patients at baseline.

### Conclusion

Both cerebral and splanchnic oxygen saturation decreased at altitude $\geq$ 5000 feet compared to baseline. In most patients, both cyanotic and non-cyanotic, cerebral oxygen saturation was preserved more than splanchnic oxygen saturation.

## Introduction

Specialized pediatric transport teams operate today as mobile intensive care units. They deliver advanced intensive care outside tertiary care centers for a wide variety of disorders using advanced monitoring equipment and skilled personnel [1–4]. Depending on distance between the referring and receiving hospital, some patients must be transported by air in helicopter or air-ambulance. Monitoring of oxygenation is important to ensure patient safety and the best possible patient outcome. Near-infrared spectroscopy (NIRS) is a noninvasive, method for monitoring of regional tissue oxygen saturation ($rSO_2$) [5]. It has been shown that cerebral oxygen monitoring with NIRS detects changes in oxygenation earlier than pulse oximetry in periods of apnea during airway surgery in pediatric anesthesia [6].

NIRS-monitoring is attractive in neonatal and pediatric practice not only because it is non-invasive, but also because the penetration of the signal into the tissue corresponds well with the anatomy of neonates, infants and children [7]. Pediatric studies have demonstrated good correlation between cerebral $rSO_2$ and jugular venous bulb saturation [8]. Anterior abdominal (splanchnic) $rSO_2$ has shown strong correlation with gastric intra-mucosal pH as well as serum lactate and systemic venous oxygen saturation ($SvO_2$) in children with congenital heart disease (CHD) [9]. In this group of patients with risk for low cardiac output, splanchnic $rSO_2$ correlated better with systemic markers of oxygenation and perfusion such as serum lactate and $SvO_2$ than did measurements over the renal bed. Multisite NIRS monitoring has been advocated to provide insight into the tissue response to different types of clinical interventions [10]. Somatic NIRS measurements may also be a better and earlier indicator of low cardiac output states than cerebral measurements, since the brain has efficient autoregulation [11].

During air transport the effects of altitude on patient oxygenation are of special importance. With increasing altitude, barometric pressure decreases and as a result also the partial pressure of oxygen. The result is lower oxygen saturations. Healthy children desaturated significantly, at times below 90%, in a hypoxic challenge test simulating the conditions of a commercial aircraft where the cabin is pressurized to 8000 feet, which equates to breathing 15% oxygen at sea level [12]. The effects of high altitude on critically ill children transported in air ambulance are not known. There are few studies of NIRS utilization in an air-transport environment and only two concerning pediatric patients [13, 14]. These studies were mainly performed in helicopters, with only 5 patients transported in air ambulance. The results suggested that cerebral oxygenation monitoring with NIRS can be used in a transport environment and that NIRS might be a useful complement to existing monitoring during inter-hospital transports.

In a previous methodological study, we found that the electronically stored NIRS data could be filtered and assessed off-line after the transport to improve reliability of the signal and thereby provide valuable post-hoc information about transport events [15].

The aim of the current study was to investigate how cerebral and splanchnic oxygen saturation in critically ill children transported in air ambulance was affected by flight $\geq$ 5000 feet. A second aim was to investigate any differences between cyanotic and non-cyanotic children in relation to cerebral and splanchnic oxygen saturation. Finally, we evaluated the variability of both cerebral and splanchnic NIRS sensors.

## Material and methods

### Ethical approval

Ethical approval for this study was provided by the Regional Ethics Review Board of Stockholm, Sweden (DNr 2013/1487-31/1 and 2016/2036-32).

### Study design and population

This was a prospective observational study, registered in the Australian New Zealand Clinical Trials Registry (ANZCTR) with registration number ACTRN12619001710112. Following written parental informed consent, 44 critically ill children scheduled for inter-hospital transport by a specialized pediatric transport team at the Pediatric Intensive Care Unit (PICU) at Astrid Lindgren Children's Hospital, Karolinska University Hospital in Stockholm, were enrolled in the study between January 2014 and January 2019 (convenience sampling). Exclusion criteria were lack of consent, participation in any other clinical research study, flights with an estimated duration less than 50 minutes and flights with a need for sea level cabin altitude. Transports were both acute and planned transfers to and from the PICU at Astrid Lindgren Children's Hospital. The team is staffed by a PICU consultant and a specialist anesthesia or intensive care registered nurse with a minimum of 3 years experience in pediatric anesthesia or pediatric intensive care [3].

### Equipment and procedures

Patients were transported in Beech Superking Air 200 and Cessna Citation II 550 air ambulances.

Standard monitoring during transport, including pulse oximetry, electrocardiographic monitoring, blood pressure measurements, body temperature, respiratory rate and evaluation with Comfort-B scale for level of sedation/comfort, was performed in all study patients and checked and noted in the study protocol before transport (base-line), during flight with cabin pressurization corresponding to $\geq$ 5000 feet and after transport [16]. The transport risk index

of physiological stability (TRIPS) score was recorded before, during transport at $\geq 5000$ feet and after transport for neonatal patients $\leq 30$ days at the time of transport [17].

All study patients were monitored with a cerebral NIRS-sensor placed over the forehead and an abdominal NIRS-sensor placed in the infra-umbilical area for cerebral and splanchnic regional oxygen saturation monitoring, $rSO_2$-C and $rSO_2$-A respectively (INVOS-5100C, Covidien, Mansfield, MA, USA). The sensors had the following dimensions: 17.25 $cm^2$ for neonates and infants and 28.8 $cm^2$ for pediatric patients. The probes had two light paths with an emitter/diode spacing of 30–40 mm and a light penetrating depth of 20–40 mm. Monitoring began at the hospital before patient transport and was continued during transfer in ground ambulance to and from the airport as well as during air ambulance transport and was finished upon arrival at the receiving hospital. Cerebral and splanchnic $rSO_2$ data were stored by the INVOS monitor during transport and extracted and analyzed off-line after the transport. The data points had a spacing of 6 seconds.

Transport personnel were instructed not to make clinical care decisions based on values presented on the INVOS monitor. Therapeutic interventions, including adjustment of fraction of inspired oxygen ($FiO_2$), were made according to the clinical judgement of the transport team. To reduce ambient light exposure, aluminum foil was used to cover the cerebral probe and the abdominal probe was covered under the patient´s clothes and blankets. The internal battery time of the INVOS 5100C is approximately 20 minutes, which made access to an external power supply necessary for both ground and air ambulance. The NIRS data was downloaded from the monitor using a Microsoft Excel 2010 spreadsheet (Microsoft Corp., Washington, DC, USA).

Prior to evaluation of the NIRS signals all zero and floor-effect values were removed. For visual inspection of NIRS signals the Savitzky–Golay algorithm of smoothing and differentiation of data by simplified least square procedures (least-squares fitting using 20 and 50 neighbors; 2nd order polynom) was applied to perform noise reduction in the signal [15, 18].

To further evaluate differences between cerebral and splanchnic oxygenation, the cerebral —splanchnic ratio ($rSO_2$-C/$rSO_2$-A) for each patient at baseline and at altitude $\geq 5000$ feet was calculated.

## Statistics

Data is presented as median and inter-quartile range. All statistical evaluation was performed on non-smoothed data. The variability for $rSO_2$-values were expressed by the coefficient of variation and compared with the Friedman's test with the Dunn's multiple comparison test. Wilcoxon signed-rank test was used for the comparison of column medians to a hypothetical value. Several independent populations were compared with Kruskal-Wallis statistics with the Dunn's multiple comparison test.

Statistics were evaluated by MS Excel (Microsoft Corporation, Redmond, Washington, USA) and Graph Pad Prism version 5.04 (Graph Pad Software Inc. San Diego, USA). All statistical tests were two sided and p values $< 0.05$ were considered to be statistically significant.

## Results

In total 44 patients were monitored. Complete cerebral regional oxygen saturation ($rSO_2$-C) and splanchnic regional oxygen saturation ($rSO_2$-A) data were obtained in 39 patients (Fig 1).

Median age was 12 days and median weight 3.55 kg. Cyanotic congenital heart malformations were present in 9 patients (23%). Mechanical ventilation was used in 12 patients (31%), 4 patients (10%) needed CPAP and 4 patients (10%) needed support with high flow nasal cannula (Table 1). Patients were categorized as cyanotic due to the presence of an intracardiac

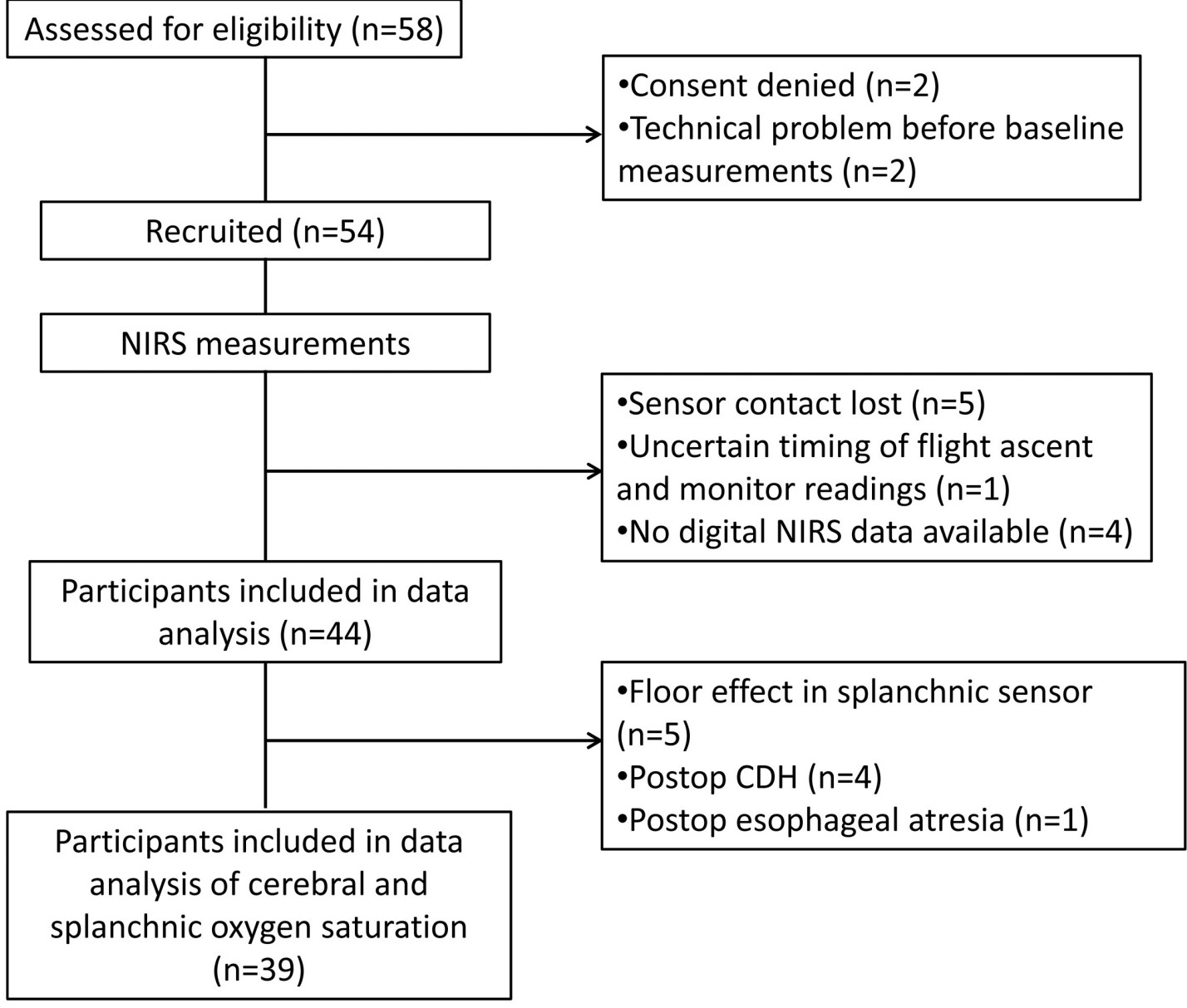

**Fig 1. Recruitment process of study participants.** NIRS = Near-infrared Spectroscopy. CDH = Congenital Diaphragmatic Hernia.

lesion or ductus arteriosus with a significant right to left shunt/mixing affecting the systemic saturation as measured by the $SpO_2$. Diagnoses that were grouped into cyanotic congenital heart malformations were: Transposition of the Great Arteries (TGA) n = 3; Pulmonary Atresia n = 2; Pulmonary Atresia & Tricuspid Atresia n = 2; Total Anomalous Pulmonary Venous Return (TAPVR) n = 1 and Truncus Arteriosis n = 1. Median pulse oximetry registrations at Take-Off were 89% (IQR 83.5–93) and 99% (IQR 96–100) for cyanotic patients and non-cyanotic patients, respectively (p<0.0001).

Simultaneous $rSO_2$-C and $rSO_2$-A values for each patient were investigated before flight (departure from hospital ward—loading into ambulance—transport in ambulance—take-off), at altitude $\geq$ 5000 feet and after flight (landing—unloading—transport in ambulance—arrival at

**Table 1. Patient characteristics.**

| Patient No. | Age (days) | Weight (kg) | Sex | Diagnosis | Cabin altitude (feet) | Barometric pressure (kPa) | Breathing support | FiO$_2$ start | Change FiO$_2$ at altitude | PiO$_2$ (kPa) | SpO$_2$% min | SpO$_2$% max | Hb (g/L) | Transport time (min) |
|---|---|---|---|---|---|---|---|---|---|---|---|---|---|---|
| 1 | 1 | 3.7 | M | Interrupted aortic arch, ASD + VSD | 6000 | 81.22 | sp | 0.21 | No | 15.7 | 87 | 94 | 175 | 70 |
| 2 | 12 | 3.3 | M | CoA + VSD | 6000 | 81.22 | CPAP | 0.21 | No | 15.7 | 99 | 100 | 148 | 70 |
| 3* | 1 | 3.8 | F | PA with intact ventricular septum | 6000 | 81.22 | sp | 0.21 | No | 15.7 | 83 | 87 | 149 | 70 |
| 4 | 5 | 5.1 | M | PPHN, biventricular failure | 5000 | 84.33 | sp | 0.21 | No | 16.4 | 85 | 98 | 182 | 75 |
| 5 | 109 | 4.8 | M | Dystrofia Myotonica type I | 6000 | 81.22 | BiPaP | 0.21 | 1 litre O$_2$ | 18.0 | 85 | 100 |  | 50 |
| 6* | 9 | 4.1 | F | PA and TA, Post op status after BT-shunt | 6000 | 81.22 | sp | 0.21 | No | 15.7 | 68 | 88 |  | 72 |
| 7* | 8 | 3.8 | M | TGA, septostomy | 6000 | 81.22 | sp | 0.21 | No | 15.7 | 75 | 82 | 151 | 75 |
| 8 | 24 | 3.7 | M | Aortic Stenosis | 7400 | 77.00 | sp | 0.21 | No | 14.8 | 97 | 100 | 150 | 81 |
| 9 | 23 | 3.5 | M | Post op status after CDH | 6900 | 78.49 | HFNC | 0.3 | 0.5 | 36.1 | 75 | 100 | 109 | 70 |
| 10* | 2 | 3.2 | M | PA and TA | 6000 | 81.22 | intubated | 0.21 | No | 15.7 | 84 | 94 | 173 | 75 |
| 11 | 1136 | 13.0 | F | Hemolytic uremic syndrome, anuria | 5800 | 81.84 | sp | 0.21 | No | 15.9 | 95 | 100 | 86 | 78 |
| 12 | 4 | 3.8 | F | Tracheomalacia | 6000 | 81.22 | sp | 0.21 | No | 15.7 | 85 | 96 |  | 99 |
| 13 | 8 | 3.2 | F | Sepsis, status after ECMO | 6000 | 81.22 | intubated | 0.38 | No | 28.5 | 98 | 100 |  | 199 |
| 14 | 12 | 4.1 | M | MAS, status after ECMO | 6000 | 81.22 | intubated | 0.46 | No | 34.5 | 90 | 94 | 142 | 60 |
| 15 | 33 | 1.4 | M | Aortic Stenosis, Left chamber hypertrophy | 6000 | 81.22 | CPAP | 0.55 | 1 | 74.9 | 90 | 95 |  | 68 |
| 16 | 3 | 3.4 | M | CoA | 6000 | 81.22 | sp | 0.21 | No | 15.7 | 86 | 99 |  | 71 |
| 17 | 39 | 1.4 | M | Post op status after commissurotomy of aorta + PDA ligation | 6000 | 81.22 | intubated | 0.45 | No | 33.7 | 93 | 100 | 109 | 76 |
| 18* | 24 | 3.2 | M | PA, Post op status after BT-shunt | 6000 | 81.22 | sp | 0.21 | No | 15.7 | 77 | 86 |  | 75 |
| 19 | 7 | 3.6 | F | CoA + VSD | 6000 | 81.22 | sp | 0.21 | No | 15.7 | 93 | 100 | 207 | 82 |
| 20 | 2 | 3.7 | M | CoA + VSD | 6000 | 81.22 | sp | 0.21 | 1 litre O$_2$ | 18.0 | 83 | 100 | 179 | 80 |
| 21 | 10 | 3.0 | M | MAS, status after ECMO | 6000 | 81.22 | HFNC | 0.3 | No | 22.5 | 90 | 100 | 109 | 70 |
| 22 | 14 | 3.9 | M | CoA + VSD | 5120 | 83.96 | sp | 0.21 | No | 16.3 | 92 | 98 | 151 | 82 |
| 23* | 5 | 2.2 | M | TAPVR + ASD | 6000 | 81.22 | intubated | 0.5 | No | 37.5 | 85 | 89 |  | 66 |
| 24 | 58 | 3.4 | M | Post op status after CDH, status after ECMO | 6000 | 81.22 | intubated | 0.4 | 0.5 | 37.5 | 86 | 100 | 124 | 207 |
| 25 | 3 | 2.5 | M | Interrupted aortic arch, ASD + VSD | 6000 | 81.22 | sp | 0.21 | No | 15.7 | 98 | 100 | 172 | 71 |
| 26 | 12 | 4.3 | F | MAS, status after ECMO | 6000 | 81.22 | sp | 0.21 | 3 litre O$_2$ | 24.0 | 79 | 100 | 105 | 75 |
| 27* | 4 | 3.9 | M | TGA, septostomy | 6000 | 81.22 | sp | 0.21 | No | 15.7 | 85 | 94 | 170 | 72 |
| 28 | 5 | 3.9 | F | PPHN, asphyxia. Status after ECMO | 6000 | 81.22 | intubated | 0.35 | No | 26.1 | 96 | 100 | 122 | 181 |
| 29* | 3 | 3.4 | M | TGA, septostomy | 6000 | 81.22 | intubated | 0.21 | No | 15.7 | 84 | 93 | 142 | 67 |
| 30 | 2 | 3.4 | M | CoA | 6000 | 81.22 | sp | 0.21 | No | 15.7 | 95 | 100 | 195 | 87 |

(*Continued*)

**Table 1.** (Continued)

| Patient No. | Age (days) | Weight (kg) | Sex | Diagnosis | Cabin altitude (feet) | Barometric pressure (kPa) | Breathing support | FiO$_2$ start | Change FiO$_2$ at altitude | PiO$_2$ (kPa) | SpO$_2$% min | SpO$_2$% max | Hb (g/L) | Transport time (min) |
|---|---|---|---|---|---|---|---|---|---|---|---|---|---|---|
| 31 | 59 | 3.4 | F | ASD + VSD with heart failure | 5900 | 81.53 | intubated | 0.25 | No | 18.8 | 96 | 100 | 112 | 60 |
| 32 | 1148 | 15.0 | F | Neck abscess | 6000 | 81.22 | sp | 0.21 | No | 15.7 | 95 | 100 | 97 | 63 |
| 33 | 9 | 4.4 | M | MAS + sepsis, status after ECMO | 6000 | 81.22 | intubated | 0.45 | No | 33.7 | 95 | 100 | 113 | 211 |
| 34 | 160 | 6.6 | M | BPD, ex premature | 7000 | 78.19 | intubated | 0.3 | 0.4 | 28.8 | 86 | 100 | 127 | 75 |
| 35 | 87 | 2.4 | M | Ex premature. Post op PDA ligation | 6000 | 81.22 | HFNC | 0.3 | 0.35 | 26.2 | 89 | 100 | | 142 |
| 36* | 3 | 3.2 | F | Truncus arteriosis | 6000 | 81.22 | CPAP | 0.21 | No | 15.7 | 86 | 95 | 196 | 84 |
| 37 | 701 | 6.9 | M | Post op status after AVSD. PPHN. | 6000 | 81.22 | HFNC | 0.7 | No | 52.4 | 96 | 100 | 98 | 72 |
| 38 | 162 | 6.3 | F | Tracheal stenosis | 6000 | 81.22 | sp | 0.21 | No | 15.7 | 93 | 100 | | 148 |
| 39 | 19 | 3.5 | M | Post op status after CDH, status after ECMO | 6000 | 81.22 | intubated | 0.3 | 0.34 | 25.5 | 89 | 100 | 127 | 156 |

FiO$_2$ = fraction of inspired oxygen, SpO$_2$ = peripheral capillary oxygen saturation, ECMO = Extracorporeal membrane oxygenation, CoA = Coarctation of the aorta, PPHN = Persistent Pulmonary Hypertension in the Newborn, MAS = Meconium Aspiration Syndrome, BPD = Bronchopulmonary dysplasia, CHD = Congenital Heart Disease, CDH = Congenital Diaphragmatic Hernia, TGA = Transposition of the Great Arteries, PDA = Patent Ductus Arteriosus, TAPVR = Total Anomalous Pulmonary Venous Return, BT-shunt = Blalock-Taussig shunt, ASD = Atrial Septal Defect, VSD = Ventricular Septal Defect, sp = Spontaneous breathing, CPAP = Continuous Positive Airway Pressure, BiPaP = Biphasic Positive Airway Pressure, HFNC = High-Flow Nasal Cannula, PA = Pulmonary Atresia, TA = Tricuspid Atresia

* = cyanotic CHD. PiO$_2$ given is with FiO$_2$ added/increased where applicable.

receiving hospital) after all zero values and floor effect values had been removed. Simultaneously registered numbers of readings for rSO$_2$-C and rSO$_2$-A before flight were 708 (IQR 521–1140), median registration time 53 minutes (IQR 43–71). At altitude ≥ 5000 feet the number of values simultaneously registered was 255 (IQR 132–435), median registration time 15 minutes (IQR 29–43) and after flight 521 (IQR 358–607), median registration time 53 minutes (IQR 41–61).

No statistically significant difference in variability was seen within rSO$_2$-C or within rSO$_2$-A when values before flight were compared to values at altitude ≥ 5000 feet and after flight. Overall, there was greater variability seen in rSO$_2$-A measurements than in rSO$_2$-C (p<0.0001), Fig 2.

Changes in rSO$_2$-C and in rSO$_2$-A between pre-flight, flight at altitude ≥ 5000 feet and after flight were investigated for each patient using all recorded values. There was a statistically significant difference (p<0.0001) and post-hoc tests for each patient and each sensor were performed (Table 2).

The data contained in Table 2 are summarized at the bottom of the table as median (IQR) for patients with and without cyanotic heart disease.

The relationship between rSO$_2$-C, rSO$_2$-A and the TRIPS score was investigated for 28 neonatal patients. Either rSO$_2$-C or rSO$_2$-A were affected in the patients (n = 10) where the TRIPS score increased between pre-transport and flight at ≥ 5000 feet (p = 0.30 and p = 0.2, respectively). Patients were grouped into three different categories with increasing pre-transport TRIPS scores 0–10, 11–20 and 21–30. The distribution of TRIPS score categories at different times is shown in Table 3. There was a decrease in TRIPS score after transport resulting in

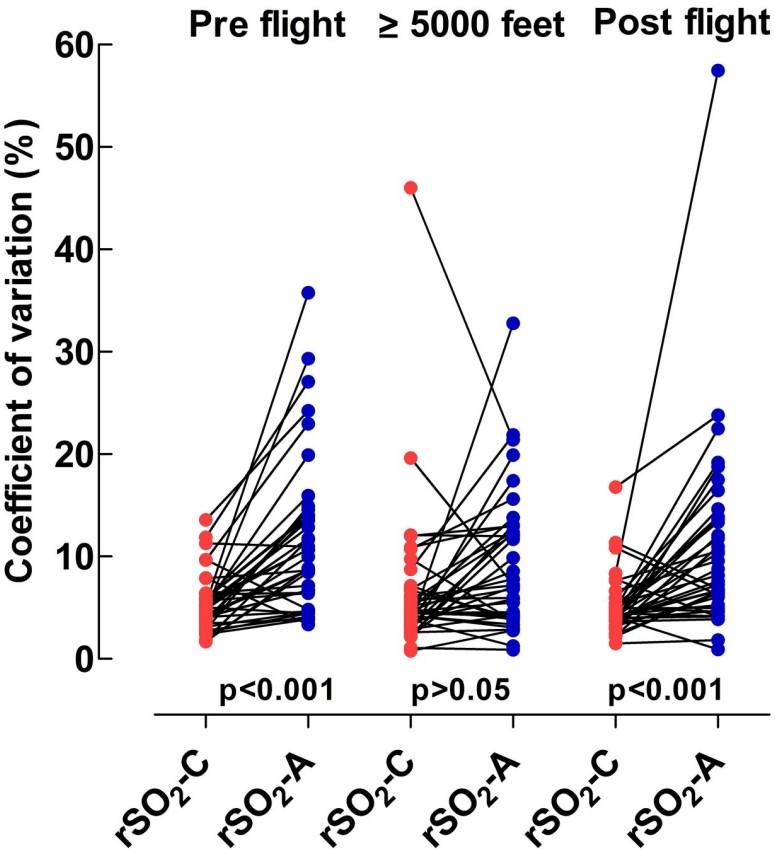

**Fig 2. Coefficient of variation demonstrating the scatter for rSO$_2$-C and rSO$_2$-A in each patient before flight, at altitude ≥ 5000 feet and after flight.** All recorded values after all zero values and floor effect values were removed are used. Red dots = rSO$_2$-C, Blue dots = rSO$_2$-A.

fewer patients in TRIPS category 21–30 compared to pre-transport. In most patients, the TRIPS score did not change during transport.

Older patients (n = 10) were monitored with Comfort-B scores for level of sedation, there was no statistically significant difference in the Comfort B levels between pre-flight, flight at altitude ≥ 5000 feet and after flight. In 22 patients (56%) rSO$_2$-C decreased at altitude ≥ 5000 feet compared to baseline, in 9 patients there was no significant difference and in 8 patients rSO$_2$-C increased at altitude. In 24 patients (61%) rSO$_2$-A decreased at altitude ≥ 5000 feet compared to baseline, in 8 patients there was no statistically significant difference and in 7 patients rSO$_2$-A increased at altitude. In patients with cyanotic heart malformations the rSO$_2$-C value at altitude ≥ 5000 feet decreased compared to baseline in 6 of 9 patients and rSO$_2$-A decreased in 4 patients. These findings are illustrated by plotting of the median values for rSO$_2$-C and rSO$_2$-A in each patient at baseline, altitude ≥ 5000 feet and after flight. Cyanotic and non-cyanotic patients are presented separately (Fig 3A–3D).

The quotients rSO$_2$-C/rSO$_2$-A were > 1 in 26 patients (67%) at baseline and in 30 patients (77%) at altitude ≥ 5000 feet. The ratio shifted from being < 1 at baseline to being > 1 at altitude in seven patients and in four patients the ratio shifted from being > 1 at baseline to being < 1 at altitude. In 25 patients (64%) the rSO$_2$-C/rSO$_2$-A ratio was greater at altitude ≥ 5000 feet than at baseline. A statistically significant difference between rSO$_2$-C/rSO$_2$-A at baseline and at altitude was found in 36 subjects (92.3% of patients).

**Table 2. Regional oxygen saturation values for all patients.**

| Patient No. | rSO₂-C (%) | | | | | | rSO₂-A (%) | | | | | |
|---|---|---|---|---|---|---|---|---|---|---|---|---|
| | Pre flight | | ≥5000 feet | | Post flight | | Pre flight | | ≥5000 feet | | Post flight | |
| 1 | 81 | (80–82) | 79 | (77–80) | 85 | (84–86) | 50 | (45–56) | 43 | (41–46) | 67 | (65–70) |
| 2 | 71 | (68–74) | 72 | (68–74) | 72 | (71–76) | 50 | (47–53) | 61 | (56–63) | 57 | (52–60) |
| 3* | 61 | (60–63) | 64 | (62–65) | 62 | (61–63) | 55 | (49–60) | 48 | (44–52) | 55 | (46–59) |
| 4 | 79 | (75–82) | 70 | (67–72) | 78 | (76–81) | 78 | (74–83) | 66 | (65–67) | 78 | (76–81) |
| 5 | 64 | (63–66) | 75 | (66–78) | 70 | (67–73) | 66 | (60–71) | 35 | (31–42) | 70 | (66–73) |
| 6* | 63 | (61–65) | 61 | (58–62) | 65 | (62–67) | 54 | (51–59) | 48 | (42–49) | 48 | (44–56) |
| 7* | 61 | (59–64) | 57 | (55–59) | 62 | (60–63) | 48 | (46–49) | 51 | (48–56) | 49 | (45–52) |
| 8 | 80 | (77–81) | 78 | (75–79) | 80 | (78–82) | 86 | (78–91) | 83 | (80–85) | 81 | (76–84) |
| 9 | 63 | (60–67) | 61 | (60–62) | 75 | (73–76) | 44 | (36–51) | 37 | (33–39) | 52 | (47–58) |
| 10* | 76 | (61–78) | 66 | (62–69) | 71 | (69–74) | 43 | (37–49) | 46 | (44–48) | 41 | (31–46) |
| 11 | 68 | (67–70) | 62 | (60–63) | 67 | (66–68) | 58 | (51–64) | 45 | (42–51) | 62 | (58–68) |
| 12 | 71 | (69–72) | 58 | (57–61) | 67 | (66–69) | 93 | (90–94) | 76 | (72–78) | 88 | (85–89) |
| 13 | 64 | (62–65) | 65 | (64–66) | 69 | (66–70) | 74 | (72–77) | 47 | (43–51) | 49 | (44–55) |
| 14 | 86 | (85–88) | 90 | (89–90) | 86 | (85–88) | 61 | (57–64) | 49 | (48–50) | 59 | (56–61) |
| 15 | 82 | (76–84) | 73 | (71–76) | 80 | (75–84) | 70 | (65–73) | 72 | (69–74) | 71 | (65–74) |
| 16 | 80 | (76–83) | 71 | (69–73) | 79 | (77–82) | 91 | (88–93) | 75 | (74–77) | 86 | (84–88) |
| 17 | 82 | (79–86) | 69 | (67–70) | 72 | (71–74) | 37 | (28–50) | 19 | (15–23) | 33 | (28–37) |
| 18* | 51 | (50–52) | 40 | (38–43) | 54 | (51–56) | 44 | (42–46) | 32 | (29–36) | 52 | (49–54) |
| 19 | 82 | (79–87) | 77 | (76–78) | 80 | (77–83) | 83 | (79–87) | 69 | (66–73) | 69 | (65–72) |
| 20 | 86 | (84–88) | 80 | (77–84) | 87 | (86–89) | 69 | (67–70) | 61 | (59–65) | 73 | (71–75) |
| 21 | 66 | (63–68) | 63 | (62–65) | 66 | (64–68) | 68 | (59–72) | 64 | (62–65) | 67 | (64–69) |
| 22 | 71 | (68–73) | 64 | (62–66) | 77 | (75–79) | 78 | (72–83) | 59 | (57–61) | 62 | (58–65) |
| 23* | 64 | (62–66) | 63 | (63–63) | 67 | (66–67) | 66 | (62–69) | 64 | (64–64) | 63 | (62–64) |
| 24 | 64 | (62–66) | 69 | (68–70) | 72 | (71–73) | 54 | (50–57) | 40 | (34–47) | 44 | (37–48) |
| 25 | 62 | (58–68) | 76 | (73–80) | 70 | (66–73) | 66 | (64–68) | 67 | (63–68) | 69 | (67–71) |
| 26 | 64 | (63–66) | 59 | (57–71) | 66 | (65–68) | 73 | (67–78) | 53 | (49–60) | 59 | (55–63) |
| 27* | 63 | (61–66) | 52 | (50–53) | 63 | (62–65) | 54 | (51–57) | 49 | (41–53) | 47 | (43–52) |
| 28 | 78 | (75–81) | 70 | (50–74) | 79 | (77–80) | 58 | (51–64) | 69 | (67–74) | 74 | (70–75) |
| 29* | 74 | (69–81) | 67 | (65–69) | 72 | (70–75) | 57 | (49–66) | 64 | (62–65) | 70 | (68–72) |
| 30 | 76 | (73–81) | 69 | (64–72) | 73 | (69–77) | 86 | (84–88) | 78 | (76–80) | 49 | (18–85) |
| 31 | 74 | (71–76) | 73 | (71–74) | 70 | (68–72) | 59 | (55–64) | 72 | (67–72) | 72 | (69–75) |
| 32 | 66 | (65–67) | 55 | (53–57) | 64 | (63–65) | 70 | (68–72) | 56 | (54–58) | 73 | (71–75) |
| 33 | 71 | (69–73) | 67 | (64–69) | 67 | (66–68) | 62 | (45–73) | 62 | (56–66) | 73 | (60–76) |
| 34 | 80 | (77–81) | 80 | (76–81) | 82 | (81–87) | 72 | (71–74) | 74 | (73–74) | 76 | (76–77) |
| 35 | 60 | (55–65) | 62 | (57–65) | 55 | (45–57) | 61 | (57–73) | 66 | (59–71) | 55 | (45–68) |
| 36* | 67 | (65–68) | 68 | (67–69) | 66 | (64–67) | 68 | (65–72) | 71 | (69–72) | 67 | (64–69) |
| 37 | 65 | (63–67) | 57 | (55–58) | 55 | (52–59) | 47 | (42–50) | 53 | (50–55) | 48 | (46–50) |
| 38 | 89 | (87–91) | 91 | (87–92) | 90 | (88–92) | 72 | (70–74) | 70 | (67–72) | 72 | (70–74) |
| 39 | 73 | (72–76) | 76 | (66–79) | 75 | (73–77) | 51 | (48–57) | 55 | (45–61) | 40 | (37–43) |
| Cyanotic | 63 | (61–71) | 63 | (55–67) | 65 | (62–69) | 54 | (46–62) | 49 | (47–64) | 52 | (48–65) |
| NON-cyanotic | 72 | (65–80) | 70 | (63–76) | 73 | (67–80) | 67 | (57–75) | 62 | (49–71) | 68 | (54–73) |

rSO₂-C = cerebral oxygen saturation. rSO₂-A = splanchnic oxygen saturation. Data are expressed as median values (IQR).

* = cyanotic CHD.

**Table 3. TRIPS score.**

| TRIPS score category | Pre-transport (n) | Take-Off (n) | Altitude ≥5000 feet (n) | Post flight (n) | At receiving hospital (n) |
|---|---|---|---|---|---|
| 0–10 | 20 | 19 | 20 | 20 | 20 |
| 11–20 | 2 | 3 | 5 | 5 | 5 |
| 21–30 | 6 | 6 | 3 | 3 | 3 |

Pre-transport TRIPS and change in TRIPS scores during and after transport. TRIPS = The transport risk index of physiological stability. n = number.

Nine patients required additional oxygen during flight. The differences and responses seen in SpO$_2$, rSO$_2$-C and rSO$_2$-A with an increase in FiO$_2$ are illustrated in Fig 4 for all nine

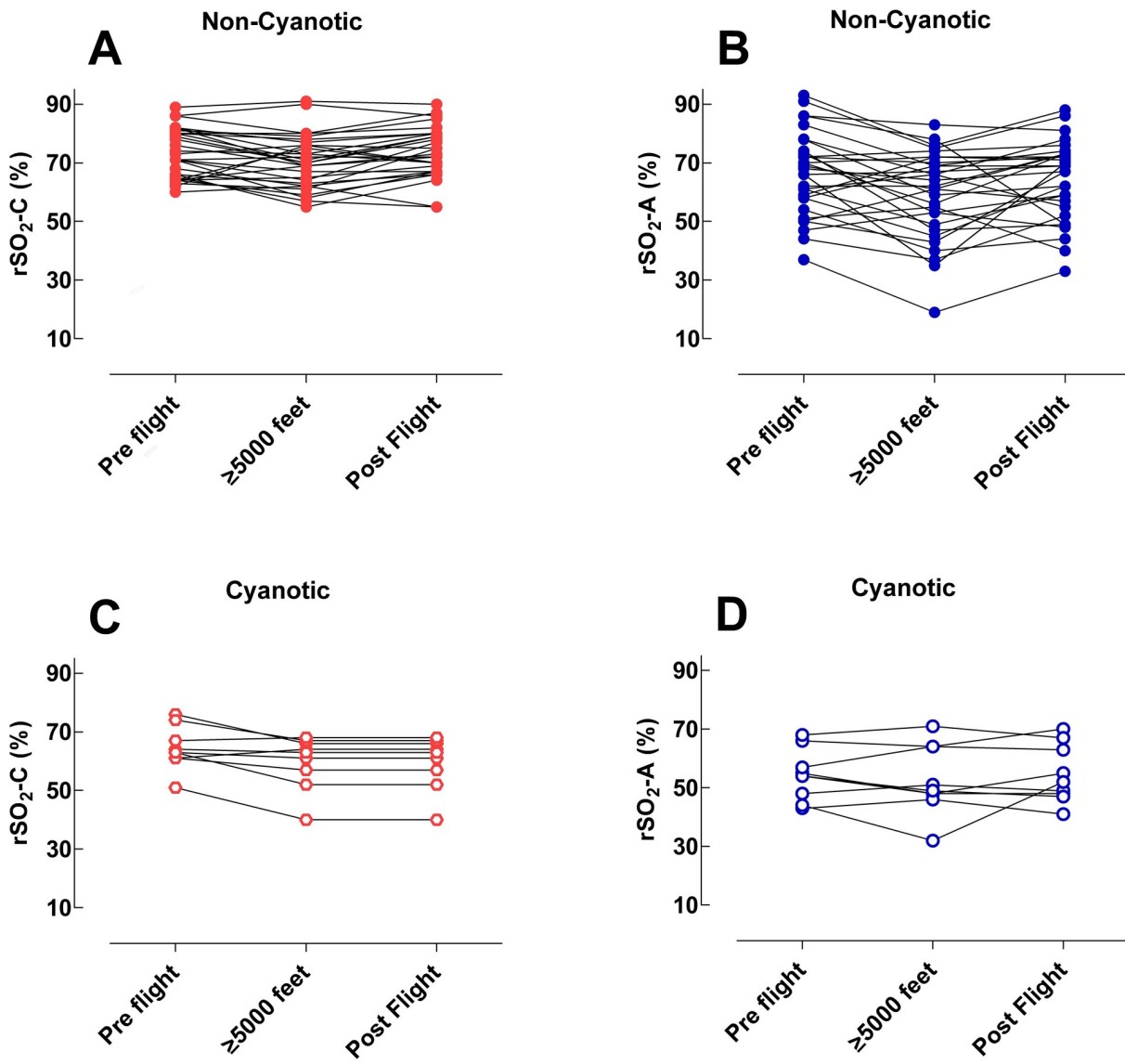

**Fig 3. The median values of rSO$_2$-C and rSO$_2$-A.** Data were derived from all recorded values, determined for rSO$_2$-C and rSO$_2$-A in each individual patient at baseline, at altitude ≥ 5000 feet and after flight. 3A: rSO$_2$-C in non-cyanotic patients, 3B: rSO$_2$-A in non-cyanotic patients, 3C: rSO$_2$-C in cyanotic patients and 3D: rSO$_2$-A in cyanotic patients.

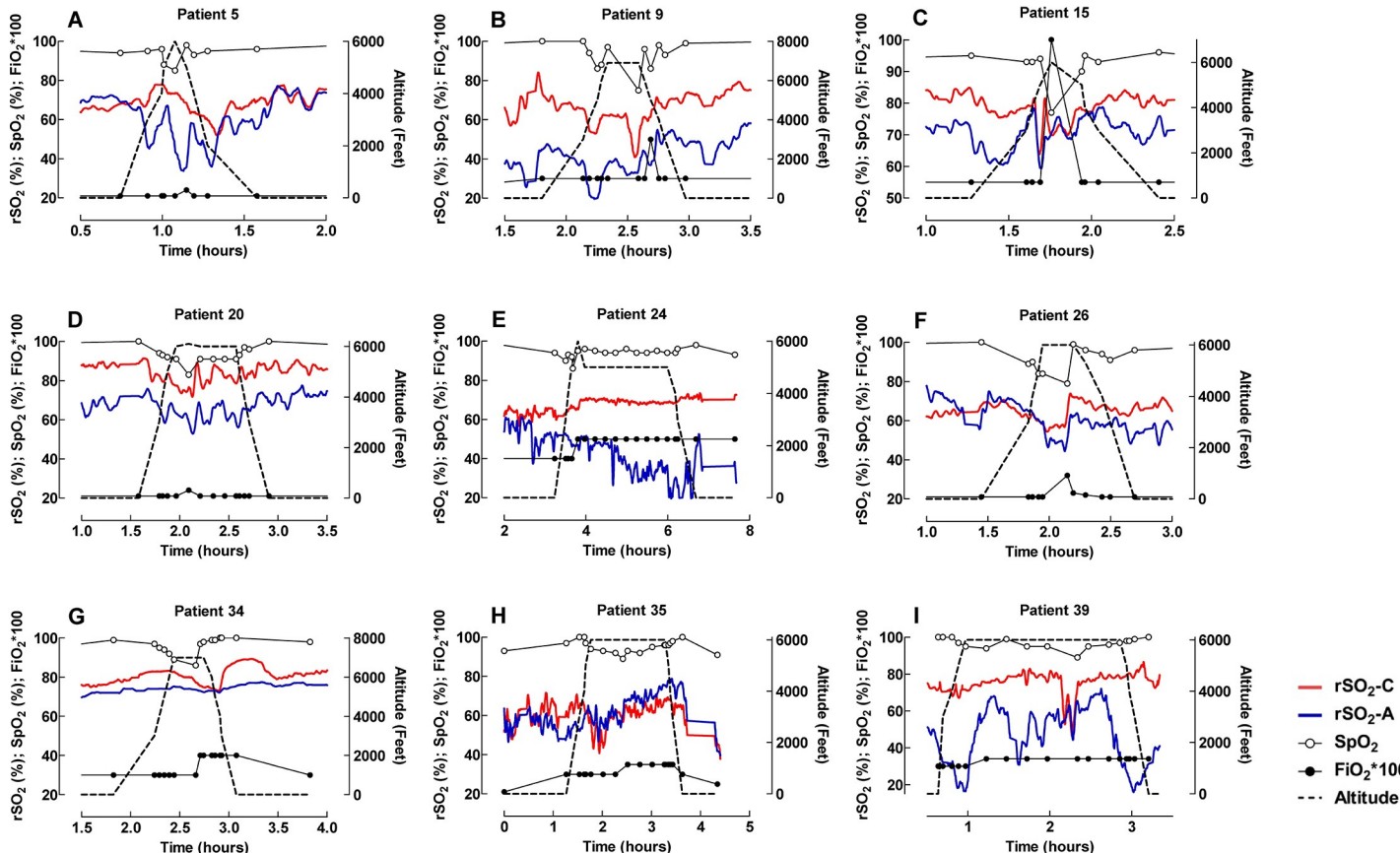

**Fig 4. Changes in SpO$_2$, rSO$_2$-C and rSO$_2$-A with an increase in FiO$_2$ illustrated for nine patients who required additional oxygen during flight.** The influence of altitude on rSO$_2$-C, rSO$_2$-A and pulse oximetry for 3 different patients was visualized after smoothing with 20 neighbours. The graphs reveal different characteristic patterns of reacting to altitude (Fig 5A–5C).

patients. In five patients: two spontaneously breathing, one on BiPaP, one on CPAP and one on HFNC, oxygen supply was changed (added or increased) during a limited time (minute to 15 minutes) of the flight (Fig 4A–4E). In the remaining four patients: three intubated patients and one patient on HFNC, oxygen was increased and continued at the higher FiO$_2$ level during the remaining part of the transport (Fig 4F–4I).

## Discussion

To the best of our knowledge this is the first study to investigate regional tissue oxygen saturation with multisite registrations from both cerebral and splanchnic areas during inter-hospital transport of critically ill children. It is also the first study which has focused on monitoring of regional tissue oxygen saturation during inter-hospital transportation of critically ill children in air ambulances.

To evaluate the consistency in measurements within each sensor throughout the entire transport event for every patient as well as between rSO$_2$-C and rSO$_2$-A sensors we studied the coefficient of variability (Fig 2). We found that the transport process per se had no influence on the variability in either sensor. Our findings of a greater variability in rSO$_2$-A when compared to rSO$_2$-C are consistent with others [19]. All patients, including those mechanically ventilated, were transported without using muscle relaxants as this is our clinical routine. The greater variability seen in rSO$_2$-A could result from the abdomen being a more mobile body

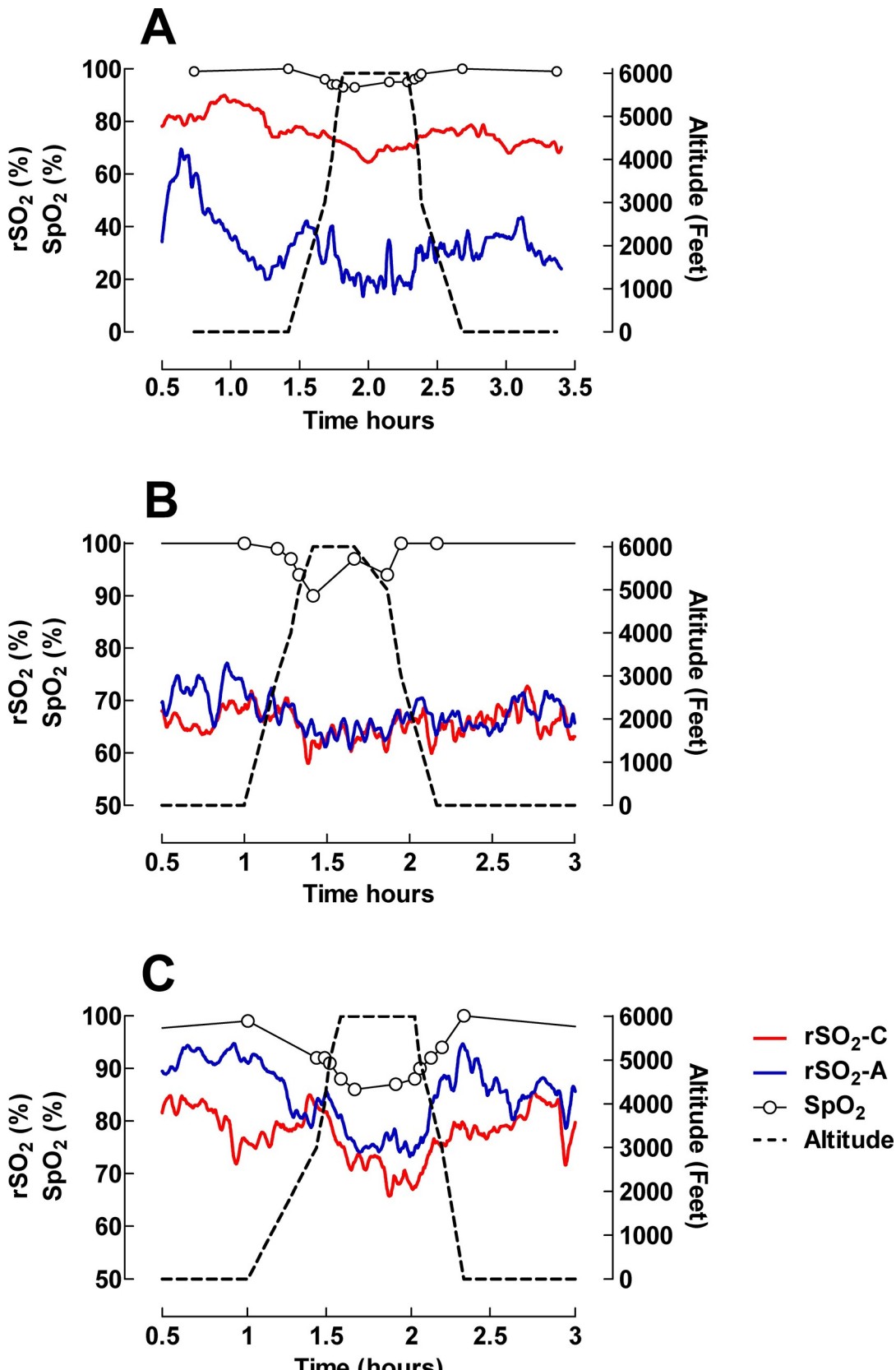

**Fig 5. Different characteristic patterns of reacting to altitude in 3 different patients.** $rSO_2$-C was higher, at the same level and lower than $rSO_2$-A in Fig 5A (Patient 17), 5B (Patient 21) and 5C (Patient 16), respectively. The NIRS curves were smoothed by the Savitzky–Golay filtering method (20 neighbours) [16].

area compared to the forehead, also in small children. Interestingly, we found that all patients excluded due to poor signal acquisition in the splanchnic sensor, had had previous abdominal surgery (Fig 1).

This study showed that in a majority of patients $rSO_2$-C and $rSO_2$-A had a statistically significant decrease at altitude $\geq$ 5000 feet compared to baseline. (Fig 3, Table 2). A large amount of data has been collected in this study and values should be interpreted and valued in a clinical perspective, since also small differences show statistical significance. Previous research has defined low $rSO_2$-C as a continuous decrease of > 20% from baseline [20]. When analyzing our data in this perspective we found a decrease of 20% in median values in only one patient. This patient (Patient 18), who belonged to the cyanotic group of patients, showed a simultaneous decrease in $SpO_2$. $rSO_2$-A values in healthy newborns were found to resemble $rSO_2$-C values by 48 h postnatal age [20]. We found 4 patients who had a 20% reduction from baseline in $rSO_2$-A, one without a simultaneous decrease in $SpO_2$. No clinical deterioration was observed in this patient during the transport, and therefore, no clinical interventions occurred. By evaluating changes in $rSO_2$-C and $rSO_2$-A in relation to $SpO_2$, we found that 6 patients showed a substantial decrease in $SpO_2$, without a corresponding decrease in $rSO_2$-C and $rSO_2$-A, interestingly 3 of these patients belonged to the cyanotic group. A more profound decrease in $rSO_2$-C and/or $rSO_2$-A than in $SpO_2$ was found in 5 patients, most obvious in $rSO_2$-A.

By using the $rSO_2$-C/$rSO_2$-A ratio we wanted to compare regional oxygen saturation of cerebral and splanchnic tissue in this cohort and investigate if a relative change could be detected at $\geq$ 5000 feet as an effect of altitude. A ratio $\geq$ 1 was seen in a majority of patients at baseline and in even more patients (77%) at altitude $\geq$ 5000 feet ($FiO_2$ was increased in only 2 patients). Among patients with a ratio < 1 at baseline, all but 2 had an elevated ratio closer to 1 at altitude. We speculate that an increasing ratio might imply that cerebral tissue was protected by auto-regulation in this increasingly hypoxic environment. We found no association between age, breathing support or diagnosis in patients with a ratio < 1. Ideally oxygen extraction would have been a more reliable measurement which could have provided additional information, but we were not able to extract continuous data from the pulse oximetry device, which limited this possibility.

We were concerned about cyanotic patients and the effects of altitude on $rSO_2$, but we found no specific patterns in $rSO_2$-C or $rSO_2$-A in relation to altitude in this group of patients, regardless of breathing support. We did note however that 7 of 9 (78%) cyanotic patients had a $rSO_2$-A value < 60% at baseline but only 10 of 30 (30%) non-cyanotic patients, and that all but one patient in the cyanotic group (Total Anomalous Pulmonary Venous Return) showed a decrease in $SpO_2$ at altitude $\geq$ 5000 feet.

We found no clear relationship between certain diagnosis and the effect of altitude. Our cohort had several patients with congenital heart disease, which could possible complicate assessment, but different reactions were also seen among other patients. This is illustrated in Fig 4 where 4A shows a spontaneously breathing patient with TGA on Prostaglandin E1 infusion and after balloon atrial septostomy, 4B a patient with high flow nasal cannula after ECMO treatment for meconium aspiration and 4C a spontaneously breathing patient with coarctation of the aorta without a patent ductus arteriosus.

Caution should be paid to meticulous application and protection of sensors in order to reduce the number of artefacts, and values should be interpreted in relation to clinical

assessments. We found that measurements of $rSO_2$-A involved more difficulties than $rSO_2$-C, such as greater variability and a higher number of artefacts, but also gave more information in addition to measurements from pulse oximetry than did $rSO_2$-C.

Limits of this study were that $SpO_2$ values from pulse oximetry were single values and not from continuous measurements which made calculations of fractional tissue oxygen extraction unreliable. The difficulties in obtaining reliable measurements on oxygen extraction is explained in more detail in supplementary materials. Infants with cyanotic congenital heart disease may have significant preductal/post-ductal $SpO_2$ differences and the lack of both pre -and post-ductal registrations with $SpO_2$ during the flight protocol is a weakness. Therefore, the potential for differences in arterial saturation across the aortic arch confounds interpretation of regional $rSO_2$ changes in these patients. Changes to NIRS readings not solely based on changes in altitude is a possible source of confounding. These changes could be due to clinical severity. A transport risk index of physiologic stability (TRIPS) score was therefore used to further group neonatal patients based on their clinical severity and the effect on $rSO_2$ during the transport. In most patients, the TRIPS score did not change during transport. Furthermore, we found no evidence that changes in $rSO_2$-readings were due to other clinical severity, as $rSO_2$-readings were not affected in patients who had a higher TRIPS score at altitude $\geq 5000$ feet than pre-transport.

In contrast the strengths of the current study were the large number of NIRS measurements collected both at baseline and during transport at altitude and our ability to remove nonsense values such as zero values and floor effect values prior to evaluation to limit the impact of artefacts on the results.

An important question on transport, where equipment needs to be limited due to space and weight, is what NIRS would add to the existing monitoring system and clinical management, and ultimately if patients' outcome would be better if NIRS is added compared to simply monitoring with pulse oximetry. Our opinion is that NIRS should be added, during transport, to those patients where the clinical management during anesthesia and intensive care would have included monitoring with NIRS in a hospital setting. Namely situations where measurements of venous saturations provide a means of titrating medications such as inotropes, vasoactive medicines and volume, or other situations with very low perfusion and oxygenation conditions in which traditional pulse oximetry could fail. To other patients, NIRS monitoring during transport is probably superfluous and would not add any clinical utility.

## Conclusions

Both cerebral and splanchnic oxygen saturation decreased at altitude $\geq 5000$ feet compared to baseline in a majority of patients. In most patients cerebral oxygen saturation was preserved more than splanchnic oxygen saturation. This was also the case for cyanotic patients even though low baseline splanchnic oxygen saturation values were observed in most cyanotic patients. The transport process per se had no influence on the variability in either sensor. NIRS-monitoring may be useful in the transport environment in the same clinical situations where it would have been used in a hospital setting. To other patients, NIRS monitoring during transport is probably superfluous and would not add any clinical utility to existing monitoring. Future studies should include equivalent continuous data for pulse oximetry for calculation of oxygen extraction to further determine the usefulness of regional oxygen saturation monitoring during transport.

## Supporting information

**S1 File.**
(DOCX)

## Author Contributions

**Conceptualization:** Tova Hannegård Hamrin.

**Formal analysis:** Tova Hannegård Hamrin, Jonas Berner, Urban Fläring, Peter J. Radell.

**Investigation:** Tova Hannegård Hamrin, Peter J. Radell.

**Methodology:** Tova Hannegård Hamrin, Staffan Eksborg, Jonas Berner, Urban Fläring, Peter J. Radell.

**Project administration:** Tova Hannegård Hamrin.

**Software:** Staffan Eksborg.

**Supervision:** Staffan Eksborg, Jonas Berner, Urban Fläring, Peter J. Radell.

**Validation:** Tova Hannegård Hamrin.

**Visualization:** Staffan Eksborg.

**Writing – original draft:** Tova Hannegård Hamrin.

**Writing – review & editing:** Tova Hannegård Hamrin, Staffan Eksborg, Jonas Berner, Urban Fläring, Peter J. Radell.

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
