## [Decision Letter · Decision Letter 0]

16 Mar 2020

PONE-D-19-35463

Influence of altitude on cerebral and splanchnic oxygen saturation in critically ill children during air ambulance transport.

PLOS ONE

Dear Dr. Hamrin

Thank you for submitting your manuscript to PLOS ONE. After careful consideration, we feel that it has merit but does not fully meet PLOS ONE’s publication criteria as it currently stands. Therefore, we invite you to submit a revised version of the manuscript that addresses the points raised during the review process.

The comments by the 2 reviewers are giving guidance towards revisions for your consideration.

We would appreciate receiving your revised manuscript by June 1, 2020. To enhance the reproducibility of your results, we recommend that if applicable you deposit your laboratory protocols in protocols.io, where a protocol can be assigned its own identifier (DOI) such that it can be cited independently in the future. For instructions see: http://journals.plos.org/plosone/s/submission-guidelines#loc-laboratory-protocols

We look forward to receiving your revised manuscript.

Kind regards,

Barbara Wilson Engelhardt, MD

Academic Editor

PLOS ONE

Journal Requirements:

Please ensure that your manuscript meets PLOS ONE's style requirements, including those for file naming. The PLOS ONE style templates can be found at http://www.plosone.org/attachments/PLOSOne_formatting_sample_main_body.pdf and http://www.plosone.org/attachments/PLOSOne_formatting_sample_title_authors_affiliations.pdf

Reviewers' comments:

Reviewer's Responses to Questions

**Comments to the Author**

1. Is the manuscript technically sound, and do the data support the conclusions?

Reviewer #1: Partly

Reviewer #2: Partly

2. Has the statistical analysis been performed appropriately and rigorously? 

Reviewer #1: Yes

Reviewer #2: I Don't Know

3. Have the authors made all data underlying the findings in their manuscript fully available?

Reviewer #1: Yes

Reviewer #2: No

4. Is the manuscript presented in an intelligible fashion and written in standard English?

Reviewer #1: Yes

Reviewer #2: Yes

5. Review Comments to the Author

Reviewer #1: Thank you for your careful data collection and thoughtful paper. I have these suggestions and questions that might guide enhancement:

1. Knowing the barometric pressure in the cabin would permit calculation of ambient PiO2. If available, these data should be included for each patient instead of, or in addition to, altitude.

2. Each patient also had pulse oximetry. Since rSO2 is a function not only of blood flow/metabolism but also of arterial saturation, it would be critical to the interpretation of rSO2 data to know what happened not only to the regional saturations, but to the differences between SpO2 and rSO2 – the best noninvasive estimation regional a-v difference, which is inversely proportional to the locoregional DO2/VO2. The authors have recognized this limitation, but I think there are ways to address this even given the lack of high-density time-synchronous measures:

In table 1, was the min/max SpO2 during transport? If so, one could calculate changes in cerebral and abdominal a-v difference using the min SpO2 and min rSO2 vs max SpO2 and max rSO2.

You have data of altitude, SpO2, and rSO2 for at least 3 patients. Are there others? Could you make the above calculations on a subgroup with the SpO2 data available? I think this would strengthen your speculations about possible patterns and mechanisms.

3. The somatic-cerebral difference or ratio is one way to ‘control’ for changes in SpO2 in patients who have normal circulatory anatomy and therefore no regional variation in SpO2, but infants with cyanotic CHD may have significant preductal/post-ductal SpO2 differences. The potential for differences in arterial saturation across the aortic arch confounds interpretation of regional rSO2 changes

4. The left-shifted fetal oxyhemoglobin dissociation curve should reduce the altitude effect on pulmonary post-capillary oxygen saturation. Therefore knowing the ambient PiO2 would help predict the pulmonary PpvO2 and SpvO2.

5. Obviously, changes in both total atmospheric pressure and the fio2 affect the PiO2. Any data to separate these effects would be helpful in interpreting an individual patient’s response to ‘altitude’ alone. This could be addressed partly by doing additional calculations in a subgroup restricted to those who did not have changes in fiO2 on transport

6. Please elaborate on the method to classify a significant change. I think you used the observed variability in each patient to calculate median and IQR, but over what time frame was variability? Was the change calculated from baseline, or from in-ambulance before takeoff, or multiply at altitude(s)? an alternative approach to examine a systematic relationship between altitude and rSO2 would be by regression, since you seem to have multiple measures of both altitude and rSO2 during ascent and descent for most if not all patients.

7. Somatic (renal, abdominal/mesentery) blood flow is highly affected by sympathetic tone. Since air transport has serious effects on patient state overall, the changes in abdominal NIRS might be expected regardless of altitude.

8. The decision to void data at at rSO2=15 or 95 is one way to deal with extreme values; the other is to leave them in. I understand that there are problems with either approach, but if a patient’s somatic rSO2 is progressively decreasing from 30’s->20’s->teens, I would not be inclined to disregard rSO2=15 as void. Likewise if cerebral rSO2 in increasing above 90, and lands pinned at 95%, I would generally interpret that as hyperemia. Exclusion of these extremes seemed to take out a significant part of the data.

9. One of the many challenges in transport, particularly air transport, would be reliability of measures. It might be helpful to present the number of minutes (or fraction of 6 second measures) during which data were not available (a value of 0 sent to the Invos data stream). It is my personal observation that NIRS measures are more reliably recovered than others during motion.

Reviewer #2: This article described a novel approach to noninvasive hemodynamic monitoring during neonatal and pediatric air ambulance transport. It has the potential to contribute to the field of neonatal/pediatric transport. The authors appropriately evaluated all patients as a whole as well as divided them into cyanotic and non-cyanotic patients, however it is suggested that they further define how they categorized patients into the cyanotic and non-cyanotic groups and what the mean/median pulse oximetry readings were for each group.

Additionally, there can be other changes to NIRS readings in patients not solely based on changes in altitude. These changes can be due to clinical severity. A transport risk index of physiologic stability (TRIPS) score could be used to further group patients based on their clinical severity and the effect on rSO2 during the transport.

The authors report statistically significant differences in both rSO2-C and rSO2-A when fiO2 was increased. It does not state in what direction these differences were appreciated. A figure would be helpful to illustrate the difference seen with increase in fiO2. Additionally, although the pulse oximetry information was not continuous data, it does seem that it correlated to the changes seen in rSO2-C and rSO2-A (figure 4). How would the addition of monitoring rSO2 add clinical utility over simply monitoring spO2? Equipment needs to be limited on transport due to space, weight and expense. The argument should be made for what NIRS would add to the clinical management of patients beyond our current monitoring system.

6. PLOS authors have the option to publish the peer review history of their article (what does this mean?). If published, this will include your full peer review and any attached files.

Reviewer #1: Yes: George M Hoffman MD

Reviewer #2: No

---

## [Author Response · Author response to Decision Letter 0]

20 May 2020

Please see enclosed "Responses to reviewers"

---

## [Decision Letter · Decision Letter 1]

5 Aug 2020

PONE-D-19-35463R1

Influence of altitude on cerebral and splanchnic oxygen saturation in critically ill children during air ambulance transport.

PLOS ONE

Dear Dr. Hannegård Hamrin,

Thank you for submitting your manuscript to PLOS ONE. After careful consideration, we feel that it has merit but does not fully meet PLOS ONE’s publication criteria as it currently stands. Therefore, we invite you to submit a revised version of the manuscript that addresses the points raised during the review process.

Thank you for substantially revising your paper and addressing most of the reviewers' concerns.Reviewer #2 suggests adding a table that summarizes the median values listed in table 2 for all patients. Rather than adding a new table, perhaps you could add the median (IQR) for those with and without cyanotic heart disease at the bottom of Table 2? This would help to summarize the data contained in the table.

We look forward to receiving your revised manuscript.

Kind regards,

Richard Bruce Mink

Academic Editor

PLOS ONE

Reviewers' comments:

Reviewer's Responses to Questions

**Comments to the Author**

1. If the authors have adequately addressed your comments raised in a previous round of review and you feel that this manuscript is now acceptable for publication, you may indicate that here to bypass the “Comments to the Author” section, enter your conflict of interest statement in the “Confidential to Editor” section, and submit your "Accept" recommendation.

Reviewer #1: (No Response)

Reviewer #2: All comments have been addressed

2. Is the manuscript technically sound, and do the data support the conclusions?

Reviewer #1: Yes

Reviewer #2: Yes

3. Has the statistical analysis been performed appropriately and rigorously? 

Reviewer #1: I Don't Know

Reviewer #2: N/A

4. Have the authors made all data underlying the findings in their manuscript fully available?

Reviewer #1: Yes

Reviewer #2: Yes

5. Is the manuscript presented in an intelligible fashion and written in standard English?

Reviewer #1: Yes

Reviewer #2: Yes

6. Review Comments to the Author

Reviewer #1: Thank you for your extensive and thoughtful revisions. Your key message, the change in cerebral and somatic rso2 at altitude, is somewhat obscured by the detail of data presented. Please consider a simple table that summarizes the median values of table 2 for all patients in the 3 conditions, and consider a simple or interaction model to show the effect of cynaotic heart disease.

Reviewer #2: The authors satisfactorily addressed the revision recommendations. The article is well written and adds to scientific knowledge of air travel in neonates.

7. PLOS authors have the option to publish the peer review history of their article (what does this mean?). If published, this will include your full peer review and any attached files.

Reviewer #1: **Yes: **George M Hoffman MD

Reviewer #2: No

---

## [Author Response · Author response to Decision Letter 1]

31 Aug 2020

We were pleased to learn that the reviewers found our manuscript much improved after the last revision. The thoughtful suggestion to add the median (IQR) for those with and without cyanotic heart disease at the bottom of Table 2 in this minor revision has been incorporated into the text as can be seen in the accompanying “Responses to Reviewers”.

---

## [Editor Report · Decision Letter 2]

3 Sep 2020

Influence of altitude on cerebral and splanchnic oxygen saturation in critically ill children during air ambulance transport.

PONE-D-19-35463R2

Dear Dr. Hannegård Hamrin,

We’re pleased to inform you that your manuscript has been judged scientifically suitable for publication and will be formally accepted for publication once it meets all outstanding technical requirements.

Kind regards,

Richard Bruce Mink

Academic Editor

PLOS ONE
---

## [Editor Report · Acceptance letter]

14 Sep 2020

PONE-D-19-35463R2 

Influence of altitude on cerebral and splanchnic oxygen saturation in critically ill children during air ambulance transport. 

Dear Dr. Hannegård Hamrin:

I'm pleased to inform you that your manuscript has been deemed suitable for publication in PLOS ONE. Congratulations! Your manuscript is now with our production department. 

Kind regards, 

on behalf of

Dr. Richard Bruce Mink 

Academic Editor

PLOS ONE